# Pooling-first Spatiotemporal Aggregation via Positional Embeddings for Efficient IVUS Segmentation

**Daewoong Ahn**[*1]                                    DWAHN@UNIST.AC.KR
[1] *Graduate School of Artificial Intelligence, Ulsan National Institute of Science and Technology (UNIST), Ulsan, 44919, Republic of Korea*

**Sangwook Kim**[*2]                                SANGWOOK.KIM@MAIL.UTORONTO.CA
[2] *Department of Medical Biophysics, University of Toronto, Toronto, M5S 1A1, Canada*

**Hyungjoo Cho**[3]                                    PHELAHAB@GMAIL.COM
[3] *Department of Applied Bioengineering, Seoul National University, Seoul, 08826, Republic of Korea*

**Soo-Jin Kang**[4]                                    SJKANG@AMC.SEOUL.KR
[4] *Department of Cardiology, University of Ulsan College of Medicine, Asan Medical Center, 88, Olympic- ro 43-gil, Songpa-gu, Seoul, Korea*

**Jimin Lee**[1,5]                                    JIMINLEE@UNIST.AC.KR
[5] *Department of Nuclear Engineering, UNIST, Ulsan, 44919, Republic of Korea*

## Abstract

Intravascular ultrasound (IVUS) segmentation often relies on 3D convolutional models to capture inter-frame context, resulting in heavy computational overhead that limits real-time clinical deployment. In this study, we propose ***Simpool***, a lightweight spatiotemporal aggregation method that replaces 3D convolutions with a GAP-first pooling strategy followed by learnable frame-wise positional embeddings which recovers temporal structure without explicit 3D operations. On a large-scale IVUS dataset of 3.7M frames across 125 cases, Simpool achieves 94.6% reduction in decoder parameters without compromising performance relative to 3D CNN baselines. Our results suggest a new direction for efficient, real-time IVUS analysis in clinical settings.

**Keywords:** Intravascular Ultrasound (IVUS), Parameter efficient segmentation

## 1. Introduction

Intravascular ultrasound (IVUS) is widely used in interventional cardiology for assessing lumen and external elastic membrane (EEM) boundaries (Xu and Lo, 2020). Recent approaches adopt 3D CNNs to capture inter-frame context in IVUS sequences (Kim et al., 2024; Nishi et al., 2021; Zhang et al., 2025; Yang et al., 2018), but incur significant computational cost due to spatiotemporal feature aggregation, limiting real-time clinical deployment. Notably, IVUS data exhibits strong temporal redundancy, where adjacent frames contain highly overlapping anatomical information, suggesting that heavy 3D modeling may yield diminishing returns.

We propose **Simpool**, a lightweight alternative that replaces 3D decoders with a pooling-first strategy: spatiotemporal features are compressed via global average pooling (GAP), then frame-specific temporal structure is recovered through learnable positional embeddings. On a large-scale dataset of 3.7M frames across 125 cases, Simpool achieves comparable segmentation performance while reducing decoder parameters by over 90%.

---

* Contributed equally

## 2. Materials and Methods

We formulate IVUS segmentation as temporal contour regression from polar images. Each input is a 3D tensor of size $1 \times T \times H \times W$ ($H = 512, W = 320$), and the model predicts lumen and EEM boundaries of the center frame as radial contour vectors with 36 angular bins each (72 outputs total).

Simpool replaces the 3D decoder head with a lightweight pooling-based alternative. Given the final feature map from an EfficientNet3D backbone (Tan and Le, 2019), we apply GAP across all spatial and temporal dimensions, project the result to a 256-dimensional embedding with SiLU activation, and add a learnable frame-wise positional embedding of size $T \times 256$ to recover temporal structure. A shared linear layer then maps each frame-conditioned embedding to 72 contour outputs.

We use a temporal IVUS dataset from Asan Medical Center (Seoul, South Korea) comprising 125 pullbacks with per-frame contour annotations (79/17/29 train/val/test; 234K/53K/80K frames). We evaluate temporal contexts $T \in \{7, 11, 15\}$ and up to $T = 64$ for Simpool. All models are trained with AdamW ($lr = 1e-4$) for 20 epochs (batch size 128) using combined Mean Squared Error (MSE) and Dice loss.

## 3. Results

As shown in Table 1 below, Simpool achieves comparable or improved performance compared to 3D CNN models across multiple temporal contexts. For example, at 11-frame input, Simpool achieves a lumen Dice score of 0.9843 and EEM Dice score of 0.9843 which is comparable to the performance of 3D baselines. Specifically, Simpool reduces the decoder parameter count from approximately 1.37M/1.81M/2.25M for the 3D baseline at T=7/11/15 to 0.119M/0.120M/0.121M for Simpool, while keeping the same backbone. Increasing the number of input frames in 3D CNNs yields marginal performance gains while significantly increasing the parameters. In contrast, Simpool maintains stable performance across varying frame sizes with minimal parameter growth. Analysis of the learned embeddings (Figure 1) shows that positional embeddings preserve temporal locality after pooling, with embedding norms peaking at the reference frame and decaying with distance.

Table 1: Test performance and decoder parameter count across temporal window sizes. Values are mean ± standard deviation over test samples. Best score per columns are bolded. DSC: Dice Score Coefficient, JI: Jaccard Index, SDSC: Surface Dice Score Coefficient.

| Model | Frames | Decoder Params (M) | Lumen DSC | Lumen JI | Lumen SDSC | EEM DSC | EEM JI | EEM SDSC |
|---|---|---|---|---|---|---|---|---|
| Baseline(3d) | 7 | 1.367544 | 0.9819 ± 0.0128 | 0.9647 ± 0.0233 | 0.9840 ± 0.0621 | 0.9842 ± 0.0176 | 0.9694 ± 0.0307 | 0.9609 ± 0.1139 |
| Baseline(3d) | 11 | 1.810200 | 0.9829 ± 0.0123 | 0.9667 ± 0.0226 | 0.9850 ± 0.0592 | 0.9844 ± 0.0180 | 0.9699 ± 0.0317 | 0.9610 ± 0.1146 |
| Baseline(3d) | 13 | 2.031528 | 0.9838 ± 0.0120 | 0.9683 ± 0.0218 | 0.9880 ± 0.0523 | 0.9845 ± 0.0178 | 0.9700 ± 0.0312 | 0.9625 ± 0.1140 |
| Baseline(3d) | 15 | 2.252856 | 0.9814 ± 0.0127 | 0.9638 ± 0.0234 | 0.9827 ± 0.0630 | 0.9842 ± 0.0186 | 0.9696 ± 0.0326 | 0.9591 ± 0.1176 |
| Baseline(3d) | 64 | – | – | – | – | – | – | – |
| Simpool(Ours) | 7 | 0.118856 | 0.9846 ± 0.0118 | 0.9700 ± 0.0215 | 0.9885 ± 0.0527 | 0.9844 ± 0.0179 | 0.9699 ± 0.0313 | 0.9608 ± 0.1148 |
| Simpool(Ours) | 11 | 0.119880 | 0.9843 ± 0.0120 | 0.9693 ± 0.0219 | 0.9857 ± 0.0610 | 0.9843 ± 0.0179 | 0.9696 ± 0.0314 | 0.9581 ± 0.1185 |
| Simpool(Ours) | 13 | 0.120392 | 0.9833 ± 0.0123 | 0.9675 ± 0.0223 | 0.9864 ± 0.0563 | 0.9832 ± 0.0179 | 0.9674 ± 0.0309 | 0.9582 ± 0.1150 |
| Simpool(Ours) | 15 | 0.120904 | 0.9833 ± 0.0119 | 0.9673 ± 0.0218 | 0.9854 ± 0.0580 | 0.9834 ± 0.0185 | 0.9678 ± 0.0321 | 0.9566 ± 0.1183 |
| Simpool(Ours) | 64 | 0.133448 | **0.9850 ± 0.0116** | **0.9708 ± 0.0212** | **0.9886 ± 0.0517** | **0.9850 ± 0.0170** | **0.9710 ± 0.0301** | **0.9626 ± 0.1110** |

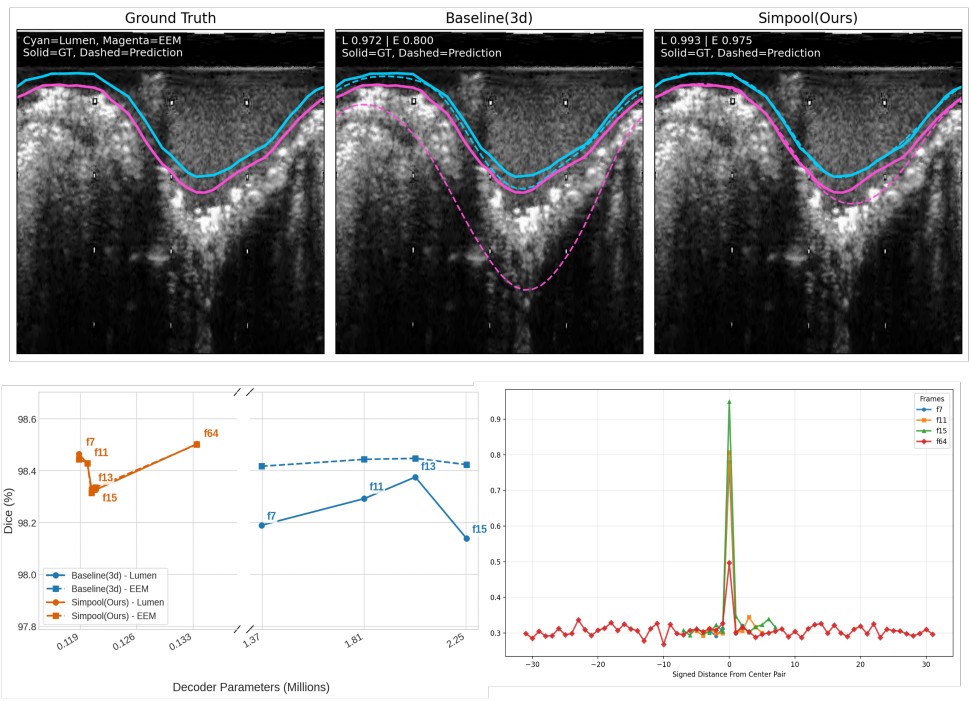

Figure 1: Overall evaluation of Simpool. (Top) Qualitative results demonstrate more accurate and stable lumen and EEM, especially in challenging regions where the baseline under-segments. (Bottom) Left: Dice-decoder parameter plot, Right: Learned positional embeddings recover temporal structure after pooling, preserving frame locality.

## 4. Discussion and Conclusion

We revisit spatiotemporal modeling in IVUS segmentation and show that effective temporal aggregation can be achieved without explicit 3D convolutions. Simpool adopts a pooling-first strategy with learnable positional embeddings, significantly reducing model complexity while preserving performance across varying temporal window sizes with minimal parameter increase. This enables efficient deployment in resource-constrained clinical settings, such as real-time IVUS analysis and interactive labeling.

In summary, Simpool provides a compact and effective alternative to 3D spatiotemporal modeling, achieving comparable performance while reducing decoder parameters by 91–95%. These findings suggest that heavy 3D aggregation may be unnecessary for IVUS segmentation and offer a practical direction for efficient medical image analysis.

## Acknowledgments

This work was supported by the National Research Foundation of Korea(NRF) grant funded by the Korea government(MSIT)(RS-2024-00344958).

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
