# OpenReview forum: "Pooling-first Spatiotemporal Aggregation via Positional Embeddings for Efficient IVUS Segmentation"
_MIDL.io/2026/Short_Papers — MIDL 2026 - Short Papers Poster_

### Official Review · Reviewer_xGAx · 2026-04-26
**Pooling-first Spatiotemporal Aggregation via Positional Embeddings for Efficient IVUS Segmentation**

**Rating:** 2
**Confidence:** 5

**Review:**

This work addresses a relevant problem in medical image analysis, namely the design of parameter-efficient models for intravascular ultrasound video segmentation, with clear implications for clinical deployment. The proposed approach, based on revisiting the decoder through global spatio-temporal pooling and positional embeddings, is interesting and enables a substantial reduction in parameters (~90%) while maintaining comparable performance.

However, the overall quality and clarity of the paper are limited by insufficient methodological details and a lack of comprehensive experimental validation. Key aspects of the architecture and processing pipeline remain unclear, and the experimental section does not fully support the claims (e.g., missing ablation study, limited evaluation). While the contribution is meaningful, its originality remains moderate, as it mainly relies on a combination of existing components.

Overall, the work has potential significance for efficient model design in medical imaging, but would benefit from improved clarity and stronger experimental evidence.

Pros:
- Relevant topic with practical implications for clinical deployment.
- Significant reduction in decoder parameters with preserved segmentation performance.

Cons:

- Limited methodological clarity and missing implementation details.
- Insufficient experimental validation (no ablation, limited benchmarking).
- Some inconsistencies and unclear elements (e.g., dataset size, figures).

**Summary:**

This study proposes an architecture dedicated to the segmentation of intravascular ultrasound videos. The main contribution lies in revisiting the decoder of a classical EfficientNet-based model to achieve comparable segmentation performance while reducing the number of parameters by approximately 90%. This is accomplished through the use of a global average pooling module applied across both spatial and temporal dimensions at the end of the backbone, combined with a learnable frame-wise positional embedding to preserve temporal structure. Experimental results demonstrate consistent performance across different temporal configurations.

**Strengths:**

- the relevance of the topic, as innovations enabling a substantial reduction in model parameters could significantly facilitate the efficient deployment of AI solutions in clinical practice.
- The substantial reduction in the number of decoder parameters while maintaining comparable Dice scores.

**Weaknesses:**

- The lack of a detailed description of the EfficientNet-3D model. In particular, more information should be provided regarding both the encoder and the decoder. The claimed reduction in decoder parameters may need to be put into perspective if the encoder remains significantly larger.
- The methodological description lacks clarity and detail. For instance, it is unclear how a 256-dimensional embedding is combined with a positional embedding of size T×256. In addition, the process by which the 72 outputs are transformed into full-resolution segmentation maps is not described. Are interpolation steps involved? If so, their potential impact on the final performance should be discussed.
- The experimental section is relatively limited. To better support the effectiveness of the proposed GAP + positional embedding design, it would be valuable to evaluate it on additional backbone architectures. An ablation study is also missing. Figure 1 (bottom left) is too small and difficult to read, and the bottom-right panel lacks sufficient explanation. Finally, the abstract states that the dataset used in this study consists of 3.7M frames, whereas the experimental section reports 234K / 53K / 80K frames for the training, validation, and test sets, respectively. This discrepancy should be clarified.

**Justification Of Rating:**

While the paper addresses a relevant problem and proposes an interesting direction for reducing model complexity, the current version suffers from significant limitations that prevent a proper assessment of its contribution. In particular, the lack of methodological clarity (e.g., architecture details, embedding integration, output reconstruction) makes the approach difficult to understand and reproduce. In addition, the experimental validation is insufficient to support the claims, with missing ablation studies and limited evaluation.

---

### Decision · Program_Chairs · 2026-05-08

Accept (Poster)